# Improvement of Disability Rights via Geographic Information Science

**Sultan Kocaman** [1],*  **and Nadire Ozdemir** [2]

1 Department of Geomatics Engineering, Hacettepe University, 06800 Ankara, Turkey
2 Department of Philosophy and Sociology of Law, Faculty of Law, Ankara University, 06590 Ankara, Turkey; naozdemir@ankara.edu.tr
* Correspondence: sultankocaman@hacettepe.edu.tr

**Abstract:** Rights, legal regulations, and practices often arise from societal and scientific developments, and societal transformations may originate from new legal regulations as well. Basic rights can be re-defined with advancements in science and technology. In such an evolutional loop, where mutual supply is obvious, combined legal and technological frameworks should be exercised and developed for practicing human rights. The main aim of this article is to propose a conceptual and methodological framework for the improvement of disability rights in the light of recent advancements in geographic information science (GIScience), in particular for those with motor disabilities, for whom questions related to "where" are essential. The concept of disability is discussed, considering different aspects, and a new methodological framework is proposed in which Geographic Information Systems (GIS), volunteered geographic information (VGI) and citizen science are at the core. In order to implement the framework at the national and international levels, a spatial data model should be developed first. The new data collection and interpretation approaches based on VGI, citizen science, and machine learning methods may help to realize equal rights for people with motor disabilities, by enabling improved access to education, health, and travel.

**Keywords:** disability; rights; GIS; VGI; citizen science; SDI; social equality

## 1. Introduction

Disability studies are interdisciplinary, utilizing insights from psychology, sociology, cultural studies, education, and more [1] (p. 5). Geography is also a discipline that can be utilized in studies in this area. Geographers have an important role in the political struggle against injustices caused by spatial formations that oppress disabled people [2] (p. 203). With the rapid development of geospatial technologies, analyzing the geographical facts and understanding societal needs have become even more achievable. The term 'Geographical Information Science' (GIScience), as discussed by several geographers [3–5], has expanded the early definitions of geographical information systems (GIS) ("*Measuring and representing geographical phenomena . . . while interacting with social structures*" [6] (p. 75)) by examining their impacts on individuals and society [5,7]. As highlighted by the Association of American Geographers [8], GIS and GIScience can contribute to public policy for several crucial issues, such as "*climate change, immigration, health, civil rights and racism, transportation, energy, electoral redistricting, natural resources, social justice, the environment, and many others*".

Sustainable development and reducing inequalities have become primary goals of all nations. The United Nations (UN) General Assembly adopted 17 Sustainable Development Goals (SDGs) to emphasize the critical problem domains for people and the environment. These aim towards ending poverty, fighting inequality, and tackling climate change [9]. Although some of the SDGs explicitly encompass geographical information, such as Climate Action (SDG 13), Sustainable Cities

and Communities (SDG 11), Life on Land (SDG 15), Clean Water and Sanitation (SDG 6), and so forth, many other SDGs and their indicators require reliable geographical data and spatial analysis methods as information synthesizers. The potential of GIScience and geospatial technologies for monitoring different SDG indicators has been studied by several researchers [5,10–17], and challenges such as the selection of appropriate cartographic representations for improved interpretation of the indicators [18] or suitability of the indicators [19] have been discussed. According to Envision2030 of the UN Department of Economic and Social Affairs [20], the SDGs explicitly include disability rights eleven times and are aimed at improving the lives of persons with disabilities. The SDGs that are particularly related to disability include, but are not limited to, Quality Education (SDG 4), Decent Work and Economic Growth (SDG 8), Reduced Inequality (SDG 10), and Sustainable Cities and Communities (SDG 11). Disability is also referenced in data collection and the monitoring of the SDGs, as described in Envision2030 [20]. Although a few studies exist on urban access inequalities [21,22], further attention from researchers and governmental institutions is required on this subject to be able to achieve the goals. Similarly, increased public awareness and participation, and collaborative work among all parties, are also essential.

Citizen science (CitSci) is defined as the active participation of non-professionals in scientific processes, voluntarily and at diverse contribution levels (e.g., data collection, interpretation, analysis, quality control, hypothesis generation, and testing, etc.) [23]. The Oxford English Dictionary recently defined citizen science as: "scientific work undertaken by members of the general public, often in collaboration with or under the direction of professional scientists and scientific institutions" [24]. Although historically, amateurs have served science and been seen as very credible, scientific works have mainly been carried out by professional scientists since the end of the 19th century [25]. With the emerging CitSci methods and platforms, amateurs have increasingly been contributing to scientific projects. This is also thanks to the availability of online educational resources, and the development of information, communication, and mobile geospatial technologies.

As a dimension of the relationship between law and geography, this article explores whether geoinformation technologies incorporated with CitSci approaches can be useful tools to improve the rights of disabled individuals. This study aims to develop a point of view on the impact of geographical features on disabled lives and rights. One of the characteristics of the geographical aspect of law is using 'where' questions in the analysis [26]. This study, by using the "where" question, aims to improve the enforcement of legal rights established in black letter law. With the assumption that disabled persons, as subjects of injustices, should be the policy shapers on disability rights. To realize this purpose, we propose to use citizen science approaches and geoinformation technologies. Here, the ideal solutions will be found via evidence-based policy-making, by addressing the problems and structuring them to find solutions collaboratively. The "when" and "how" questions were therefore integrated into the methodological framework proposed in this study since persons with motor disabilities especially suffer from environmental or physical and social challenges to mobility; for instance, changing their position or moving to target locations to exercise their rights.

In the following sections, the concept of disability is described in a conceptual legal framework that considers the long-term relationship between law and geography, the structural injustice caused by geographies, and the United Nations' efforts on the rights of persons with disabilities. A methodological framework based on geospatial technologies and citizen involvement is also proposed. In the methodological framework, individual capability based on different kinds of motor disabilities and potential assistance that could facilitate mobility (e.g., vehicles, wheelchairs, specially designed apps, social support, etc.) are considered, the importance of establishing a disability-related spatial data infrastructure (SDI) is emphasized, and both CitSci and machine learning (ML) methods are proposed for understanding and analyzing social and physical enablers and disablers.

## 2. Conceptual Framework

### 2.1. Law and Geography

With their "ubiquitous nature", both geography and law, as established disciplines, examine the controlling forces on things and persons [27]. The relationship between these two disciplines has often been the subject of legal studies. Studies on law and geography have different aspects depending on how the connection between these two disciplines is established or defined. Sometimes they interplay, sometimes they try to dominate one another, or sometimes they benefit from each other; this article focuses on this aspect.

The interplay between these two disciplines has long been studied. Montesquieu and Eugen Ehrlich, as pioneers in this area, explained the impact of geography on legal rules and institutions [28]. For example, Montesquieu, in The Spirit of Laws, explained the impact of climate on laws or governmental structures [29]. Contemporary studies have carried this interaction to different dimensions. For example, Oñati International in the Law and Society series (by the editorship of William Taylor) collected essays that explore the relationship between landscapes or spaces and individual identity, framed by law [30]. This relationship has been considered from the perspectives of different aspects, according to the researcher's own field. Rutherford Platt, who defines himself as both a lawyer and geographer, explored the 'influence of law over the human use of land' [31] (p. 7). In his book Land Use and Society, he explains this interaction as follows:

> ( . . . ) we may note that law is both a dependent variable, shaped by the real world of the geographer, and an independent variable that itself shapes the human environment in sometimes unexpected ways." [31] (pp. 42–43).

Lauren Benton studied one of these unexpected interactions through the concept of sovereign. By tracing seas, oceans, or upriver regions, he tried to explore the interaction of law and geography that shaped sovereignty in history [32]. Similarly, Victor Prescott and Gillian D. Triggs studied the politics of geography on boundaries in their book *International Frontiers and Boundaries: Law, Politics and Geography* [33]. Melissa Tatum and Jill Kappus Shaw examined the intersection of law, culture, and environment on America's public lands [34]. In fact, studies on environmental law, land law, and even space law can all be considered as relevant to this mutual relationship.

Within this mutual relationship between law and geography, sometimes one dominates the other. For instance, humans shape geography by using law as a legitimate tool. Sometimes, geography shapes law and society as an uncontrolled force. Law does change form and content in different places in the world due to geographical traits. In this sense, how law functions in any society require a geographical analysis [26] (p. 510), or geography can be a tool to understand the enforcement of a particular human right [26] (p. 512). For an analysis of the geopolitics of hunger (or the right to food), see Rienner's work [35].

The interplay between law and geography has been the subject of many interdisciplinary research projects. Irus Braverman and her colleagues conceptualized this special interdisciplinary area as "legal geography" [36]:

> "Legal geography is a stream of scholarship that makes the interconnections between law and spatiality, and especially their reciprocal construction, into core objects of inquiry. ( . . . ) Legal geographers note that nearly every aspect of law is located, takes place, is in motion, or has some spatial frame of reference. In other words, law is always "worlded" in some way ( . . . ) Legal geography is not a sub-discipline of human geography, nor does it name an area of specialized legal scholarship. Rather, it refers to a truly interdisciplinary intellectual project."

As a legal geography study, this article aims to investigate how geographical structures can shape structural injustice in disabled lives, and how technology can be a useful tool to remedy this injustice. It is believed that social and legal norms in a given society shape the perception and responses to disability. Besides social and legal grounds, geography is also crucial in enabling the rights of disabled persons, and technology as a tool for the improvement of disability rights.

## 2.2. Disability: A Difficult Concept to Define

This study, focusing on disability rights, should have conceptual clarity on what disability is from the outset. However, as a complex and multi-faced concept, disability is hard to define, and definitions vary according to and depending on the different conceptual frameworks [37] (pp. 5–7). When we look at the literature, we can find at least five different approaches to disability; each defines the concept differently, and their requested social policy differs consequently.

The first approach is called the "*medical model*", and it emerged around the 18th century [38] (p. 25). It sees disability as a personal problem caused by health issues to be improved through medical methods [37] (p. 9). In this traditional view, disability is medicalized and individualized [39] (pp. 17–18). As a response to the inadequacies of this view [38] (p. 28), a second approach, called the "*social model*", emphasized that disability cannot be oversimplified by medical explanations. According to the social model, inadequacies of social and environmental conditions construct disability [37] (pp. 9–11). This model also focuses on the difference between impairment (which is individual and biological) and disability (which is socially constructed) [39] (pp. 34–35). For more information on these two main concepts, please see Thomas's work [40]. A third approach, called the "*relative model*", as a combination of the previous two models claims that disability cannot be explained only by medical factors, but social and environmental factors also play a role [37] (p. 11). A fourth model does not see disability as "imperfect or limiting in life", but it accepts disabled people as having a *different lifestyle* that can be considered as a minority [37] (p. 12). For example, people in the deaf community see themselves as cultural minorities because they have a different language (sign language) that the majority do not use [37] (p. 12). For more information about the minority model, see Mitchell and Sharon's work [41]. Finally, there is the (Nordic) relational approach, which defines disability as a "*mismatch*" between the individual and the environment [39] (p. 25). It defines disability as a consequence of the relationship between the person and their environment [1] (p. 20).

These five different approaches differ in definition, and this also shapes the policies they suggest. As our article will be based on the relationship between geography and disabled persons, we accept the interaction between legal rights and the physical environment. We subscribe to the idea that accessibility (of legal rights) depends on both environmental conditions and individual capability [42]. In this regard, our approach is more similar to the "*relative model*" explained above. We accept that disability can be explained both medically and socially or environmentally. By recognizing the role of legal geography in shaping "social and material realities" [43], we will try to bring insight into how structural injustice can be caused by the physical world, and how geoinformation can be used to eliminate injustice and enable disability rights.

## 2.3. Structural Injustice: How (Legal) Geography Dis/Enables

Brendan Gleeson questioned how spatial issues can disable rather than enable people with physical impairments [2] (p. 1). Gleeson used the term "enabling geography" to state the ways in which the geography discipline might play "an enabling role" in disabled persons' access to justice [2] (p. 195). He emphasized that socio-spatial analysis gives clues about the position of disabled people in society [2] (p. 59). This position is beyond black letter laws and exists in the real lives of disabled persons.

Non-inclusive or non-adequate planning makes disabled rights inaccessible, even though they are declared on black letter laws [44] (pp. 5–6). For instance, disabled persons have the right to enjoy healthy environments. However, considering the geographical features of parks or recreation areas, we can often question access to these spaces by disabled persons. They have the right to justice, however, the physical features of many courts make them difficult to access independently, and this creates an obstacle for access to justice [45] (p. 95). Additionally, the physical conditions of courts can obstruct disabled lawyers' right to work [46]. As Malloy addresses, mobility impairment is understood as a civil rights matter, instead of seeing its connection with land use and planning issues [44] (p. 6). Disability rights in terms of the black letter law cannot be used, as structural injustices are obstacles to this. Structural injustice exists within the structures of the physical world, leading disabled persons to become non-users of their legal rights. Iris Marion Young explains structural injustice as follows [47] (p. 52):

> *"Structural injustice exists when social processes put large groups of persons under systematic threat of domination or deprivation of the means to develop and exercise their capacities, at the same time that these processes enable others to dominate or to have a wide range of opportunities for developing and exercising capacities available to them."*

Physical restrictions and obstacles are everywhere to limit disabled people's accessibility to public places [48] (p. 279). Similarly, street furniture is suitable for neither blind people nor those with a wheelchair [48] (p. 279). In this sense, all these physical obstacles restrict disabled peoples' access to many places or even shape their preferences on where to go [48] (p. 279). Thus, geographical structures become structures of injustice for disabled persons. Most architects do not have enough knowledge of impairment for their designs to take into account the needs of people with impairments [48] (p. 279). Lack of knowledge of impairment relates to an important issue in disability rights, the right holder's experience is crucial in the struggle against injustices.

There are many disability types, and this makes the experience of each disabled person unique. In this regard, we can say that the structural injustice each disabled person faces also varies. This highlights the importance of disabled people being the subject of policy-making in disability. Robert Drake has highlighted this issue by stating that it is "inappropriate for non-disabled people to speak on behalf of disabled people" [49]. For the same reasons, he sees it as "inappropriate for non-disabled people to do research about disabled people" [49]. Similar views shared by Tom Shakespeare state that disabled people should be at the core of the politics of disability and should be the decision-makers on disability, rather than non-disabled people [39] (p. 185). Here, we propose a methodological framework that can enable this.

### 2.4. The Convention on the Rights of Persons with Disabilities

Adopted by the United Nations General Assembly in 2006, The Convention on the Rights of Persons with Disabilities (the Convention) aims to change the perception towards disabled persons. It views persons with disabilities as right holders, instead of being the object of "charity, medical treatment, and social protection" [50]. The Convention declares the basic human rights of disabled persons and calls for state parties to "promote and protect the human rights of all persons with disabilities" (Preamble j). It specifically highlights the importance of disabled persons being "actively involved in decision-making processes about policies" that concern them (Preamble o).

"Accessibility" is an important issue that is emphasized by the Convention. It ensures that states take appropriate measures to enable disabled persons to achieve full participation and independent life. Within accessibility, it mentions "physical environment", "transportation", "information and communications" (technologies and systems), and all other facilities and services in urban or rural places (Article 9).

The Convention guarantees that states should "provide accessible information to persons with disabilities about mobility aids, devices, and assistive technologies, including new technologies, as well as other forms of assistance, support services, and facilities" (General Obligations h). The Convention also ensures other basic rights and freedoms, such as access to justice (Article 13), living independently and being included in the community (Article 19), personal mobility (Article 20), respect for privacy (Article 22), education (Article 24), health (Article 25), work and employment (Article 27), and participation in political, public and cultural life, and recreation leisure and sport (Article 29–30).

In short, the Convention defines disabled persons as right holders and calls on state parties to guarantee the basic rights and freedoms of disabled persons, which sometimes require affirmative action programs in order to ensure full equality. Of course, all these rights and freedoms are abstract on paper, however, can be realized within actual life by technological tools.

## 3. Methodological Framework to Improve Disability Rights

Inclined roads, cobblestone pavements, a lack of elevators, and any other physical obstacles for disabled chairs can make it difficult to exercise abstract legal rights [42]. In order to determine these

obstacles, the features of a given space can be identified using geospatial technologies, with the help of CitSci. This geographical aspect will help to describe the mutual relationship between law and space that will eventually suggest potential improvements in disabled lives and rights. As disability conditions differ from one person to another, each disabled person's individual experience of these structural injustices is valuable. However, as stated in the Convention on the Rights of Persons with Disabilities, appropriate measures should be taken in order to enable disabled persons to live independently [50]. Stuart Blume has advocated for the use of technology for disability [51], however, he emphasized devices that disabled people can use to integrate into society. Kwan [52] analyzed the neighborhood's effect on people's behaviors and outcomes and emphasized the importance of time and people's exposure to environmental influences in human mobility. The study concluded that analytical methods and modeling approaches are needed to assess exposure, which can be based on high-resolution space-time trajectories collected by location-aware geospatial technologies.

Our proposed framework involves a methodological approach to identify the potential obstacles in daily life that hinder movement for people with motor disabilities and suggests tools and methods to overcome them or to assist people by also integrating the time dimension so that they can exercise their rights just like any person without disabilities. When realized, such systems would support the reduction of social inequalities and lead to a fairer societal state. In order to emphasize the essentiality of the development of proper policies and the establishment of relevant infrastructure, a systems approach is followed, and special attention is paid to the importance of the development of a motor-disability oriented SDI and its core elements. The following sections include descriptions of the proposed methodological framework and an analysis of the contribution of citizen science to this framework.

## 3.1. Use of Geoinformation Tools and Technologies as Enablers for Motor Disability

Geospatial technologies are widely used currently, thanks to mobile geolocation sensors and services, and online data-sharing possibilities. Geoinformation technologies have traditionally been used in applications such as mapping, urban and rural planning, and construction, but nowadays the application areas have been expanded to solving modern problems such as autonomous driving, mobile navigation (indoor and outdoor), smart buildings, cultural heritage documentation, advanced environmental modeling applications, healthcare, and smart city applications [53–62]. A systematic approach for using geospatial technologies for the purpose of improving disability rights and lives must include structuring the problem by identifying the basic requirements, designing a system at the national level that can also be applied internationally, and developing a living system which can be updated with novel technologies and user inputs.

As a first step, to analyze the system requirements, a generic workflow to answer the where-when-how questions for individual capabilities is provided in Figure 1. In other words, the problem can be seen as a navigation issue between points A–B, and an analysis of functional requirements can be made. To start with, the disability types and available assistance determine the individual capability for the purpose of movement (location change) at a given time. The where and when questions basically involve the spatio-temporal elements: the start-end points (A–B), time (or temporal frequency) of mobilization, the geographical structures (entities) that exist in the area (e.g., transportation infrastructure, city furniture, etc.), and any social or authoritative regulations (e.g., operational restrictions). With regard to the how question, one should consider the transportation options, alternative routes, potential obstacles, and also enablers which can help provide a solution for the defined capability. The spatio-temporal and social elements reflect the physical and the social environment, and improvements to these environments are possible with proper feedback after performance assessment of the solution (output of the how question). The feedbacks would also support the evidence-based policy-making. The main goal is to reduce disablers and increase accessibility, and thus the requirement for social enablers (supporters) can also be reduced, which is possible with the proper design of the environment for individuals with motor-disabilities. Ideally, by achieving this goal, the "Accessibility" guarantee of the UN Convention on

the Rights of Persons with Disabilities (the Convention) can also be achieved, and disabled persons can enjoy full and independent participation in social life.

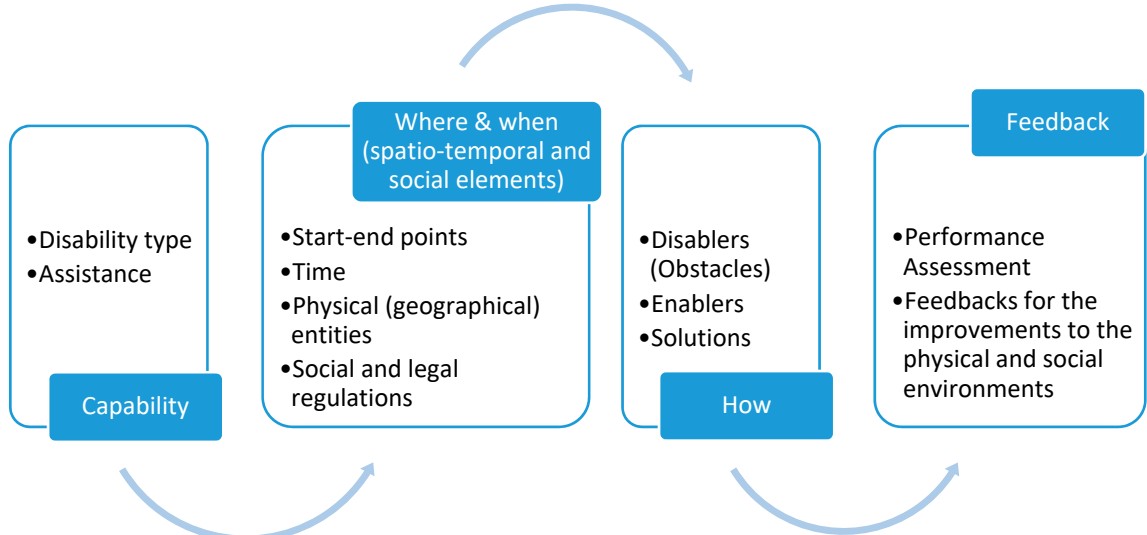

**Figure 1.** The basic elements of the where-when-how questions demonstrated in a functional scenario-based on individual capability.

An example of the where-when-how model presented in Figure 1 can be given from a lawyer or any layperson in a wheelchair who has a hearing at the Courthouse. This person needs to reach the hearing punctually. If the road from home to the Courthouse is full with physical or social obstacles for the wheelchair (for example if the sidewalk is not suitable for the wheelchairs or the park on the road is closed for any wheeled vehicles), or the Court building has no elevator for the wheelchair, or the Court building itself has some legal obstacles such as prohibiting wheelchairs then this person's access to justice would be undermined. A similar example can be given for a blind person with an assistance dog. If the Court building is prohibited for any pets, then this person will have difficulties in using their legal rights. Or if any public park were closed for pets, then this person would have difficulty in exercising their right to the environment, which they have it on paper. In all these examples, the physical and social obstacles (disablers) and enablers need to be identified in a specifically designed mobile navigation algorithm to ensure mobility. Such a navigation app can also propose different solutions based on the available assistance (i.e., wheelchair, assistance dog, a car for the disabled person, etc.) and even provide additional assistance by specially designated voice guiding for a blind person. Feedbacks can be collected in the app as well and provided to the authorities.

At the core of the system design, an enabling SDI must take place to achieve interoperable solutions at the national and international levels. An SDI is essential for facilitating the spatial data exchange between stakeholders and the community [63]. It comprises policies, standards, procedures, and access networks to enable efficient use, management, and production of geospatial data [63,64]. Some of the benefits of developing an SDI are improved access to data, reduced duplication of effort in collecting and maintaining data, better availability of data, and interoperability between datasets. Although SDI development is an accepted task for most organizations and nations, and examples exist for various problem domains such as topographic mapping, indoor, health, marine, arctic, and so forth [65–73], attempts at developing data models for the improvement of disability rights are new within the literature [74]. As explained by Rajabifard et al. [63], an SDI can be an enabling platform for sustainable development challenges. Scott and Rajabifard [11] stated that the design of any SDI requires analyzing the nature and the components of the problem, the impact of global drivers, and the needs of the user community. Our understanding of an enabling SDI, as depicted in Figure 2, is one that is a connecting and facilitating concept for several components in a highly dynamic environment that

includes people and their ever-changing needs, data both as a requirement and as a design utility for data-driven systems, standards for interoperability, dynamic policy updates, and methods and access networks that provide the necessary technological and methodological infrastructure. Here, the people, the needs of society and the environment, and the data availability, quality, and requirements are the intuitive drivers of the whole system. The policies and the standards can be seen as the bond between all parts, whereas the methods and the access networks (e.g., portals, data warehouses, etc.) form the backbone of the execution.

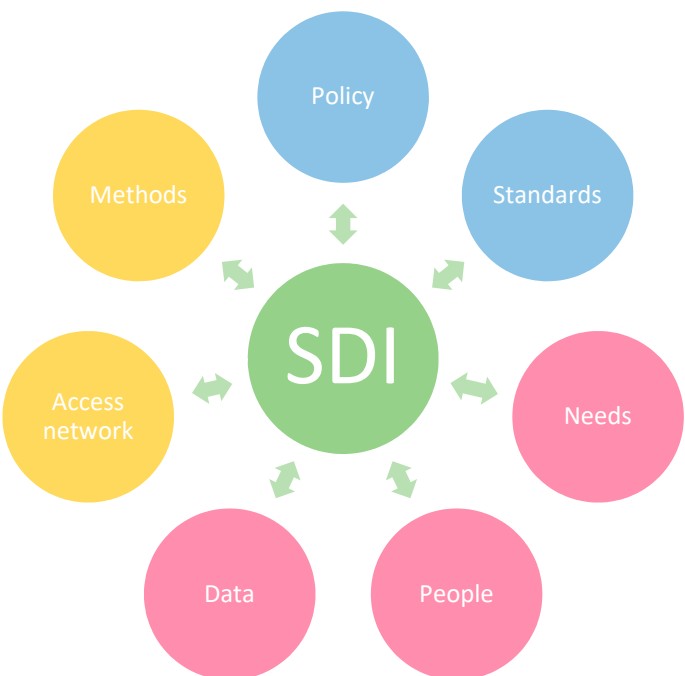

**Figure 2.** Nature and relationships between spatial data infrastructure (SDI) components (modified from Rajabifard et al.'s work [63]).

A methodological framework was designed and proposed here (Figure 3), based on the recent technological developments in geospatial data collection and the processing domain. The main design components involve identification of the SDI elements and developing a spatial data model, data collection methods, building the warehouses/repositories, accessibility of the system, data dissemination, and feedback for policy development, decision-making, and legislation.

Identification of geographical entities and their attributes, and the social elements that cause motor disability, is the first stage of the data model development. Any potential enablers and disablers (obstacles) must be included for specific disability types. The developed data model should be compatible with standards, which may need to be developed for this purpose, and should include the access methods as well. Interoperability must be ensured at this stage so that the same data model and methods can be utilized across countries, and data exchange can be made possible between different data sources, systems, and organizations. Interoperability is crucial, especially for the practice of travel rights to any destination.

The developed data model and methods need to be integrated into motor-disability related warehouses, which need to be part of the public data infrastructure and national SDIs for mapping. An integrated information repository will also serve the development of assistance tools as enablers for motor disabilities, and reinforce urban plans and planning principles. The most important output of such repositories is the facilitation of solid principles for the realization of abstract legal rights and freedoms for disabled persons, which shall be inputs for policymakers.

Acquisition of accurate, reliable, and up-to-date data is essential for the success of any SDI. The data required to establish the geographical entities and to detect the physical enablers and disablers

(as mentioned in Figure 1), both indoor and outdoor, can be collected via different methods. Several novel approaches for data collection, such as high-resolution remote sensing platforms, autonomous sensors technologies with the support of the IoT, crowdsourced big data, volunteered geographical information (VGI), social networking sites, and so forth need to be harmonized with the help of mobile technologies and ML methods. Access and analysis methods also have a special role here, as they need to provide rapid, timely, meaningful, and accurate results, and increase the functionality of the system.

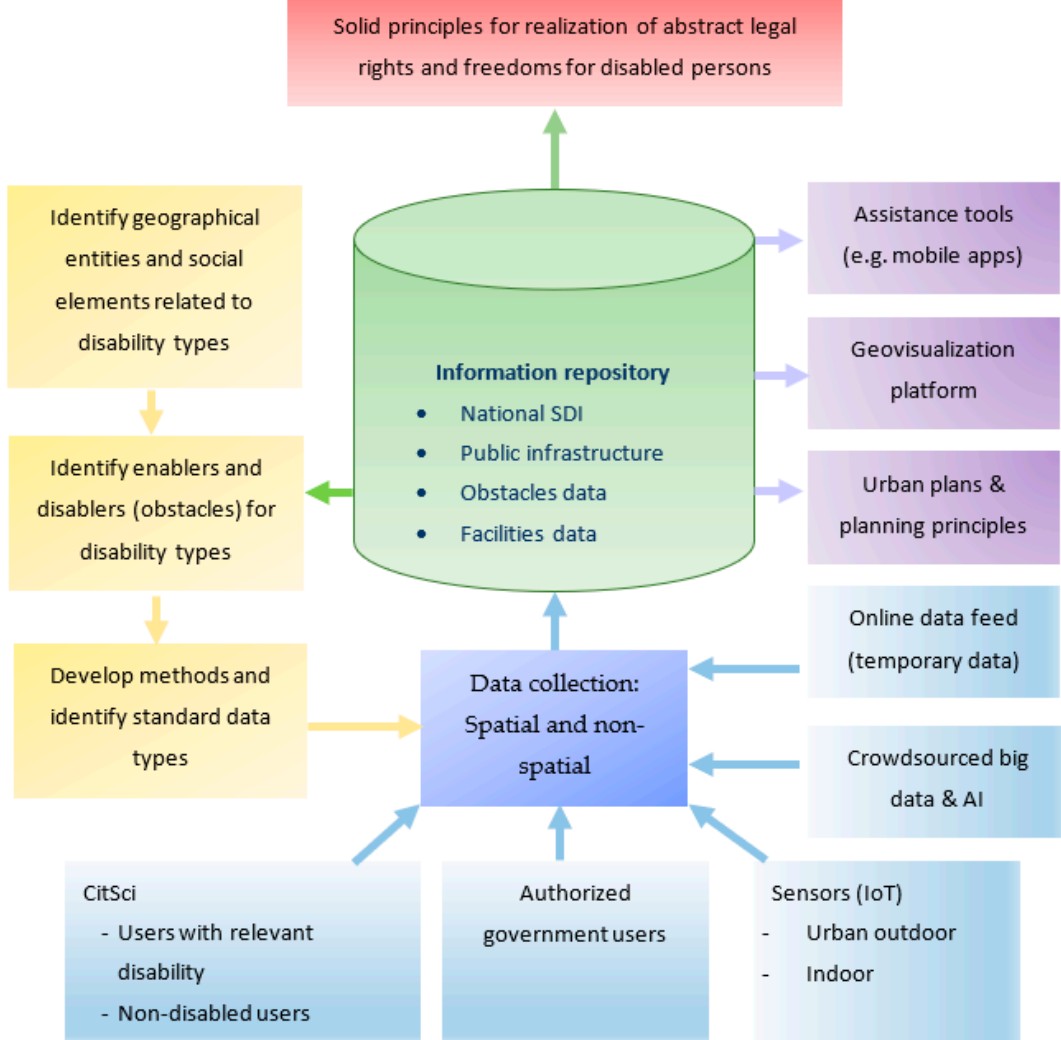

**Figure 3.** A methodological framework for the establishment of the SDI elements, which aims at improving the rules and regulations for disability rights.

As an example, mobile apps can help disabled persons get information about specific public places concerning their disability problems. A specially designed app can be useful to collect data about a specific building or a parking place depending on other disabled persons' experiences. In this regard, they would know whether/not or how to get to that place safely. However, contributions of non-disabled people or automated methods are also needed, since they can provide data about places with low or no accessibility for disabled people. On the other hand, crowdsourced data from smartphones, e.g., street images, or IoT (e.g., street cameras) with geolocation information can be used to detect potential physical obstacles automatically and possibly in real-time, which can contribute to the information repository as depicted in Figure 3. Various studies for street object detection from mono or stereo images or lidar data can be found in recent literature (e.g., [75–79]). VGI can as well be used for similar purposes, i.e., spatial data provision or accessibility validation, with the direct or indirect inputs from humans (e.g., please

see [80–82]). By using the collected big data, typologies of urban areas for persons with impairments can be derived and specific metrics can be developed (e.g., please see [83]). Here, perceptions of disabled people on the physical environment also play a key role, and naturalistic studies can be carried out to analyze the behavior of disabled people in real-world conditions (e.g., please see [84,85]).

Considering these requirements, ML and artificial intelligence (AI) methods deserve special attention. These methods can support GIS for automated processing and analysis. An AI-supported GIS platform, which employs web and mobile technologies, spatial databases, statistical and image processing software libraries, and visualization tools to assist the volunteers for spatial data collection was developed by [86]. Since the quality of data collected by volunteers is often questioned in CitSci projects (mainly due to the motivation and knowledge of the participants), ML techniques can be employed for quality assessment. A study by Can et al. [87] has shown that ML and AI techniques can support automatic quality control and thus increase the system reliability by auto-detection of VGI and CitSci collected image data contents using Convolutional Neural Networks (CNN). Another study by Foody et al. [17] has addressed the detection of slavery sites using remote sensing data and CNN. The number of deep learning studies, which aim at detection and semantic labeling of street objects by processing big data, has been increasing (e.g., [75,77,88,89]), and such approaches can as well support instant mapping of disabling physical entities on the transportation routes of disabled persons.

On the other hand, social networking sites are gaining increased attention for VGI data collection and analysis (e.g., please see [90–92]). However, a study by Yalcin et al. [93] states that using specially designed apps for CitSci data collection can provide structured data with correct position information since the users can be trained in advance. Thus the data processing can be faster, and designing an appropriate app to achieve the specific goal is also possible. Application interface design and the use of proper human-computer interaction methods are quite important also for people with special needs. Geospatial technology also offers different possibilities in this respect, such as Virtual Reality and Augment Reality. As stated by [94], the demand for interactive experiences grows by different user groups (e.g., government decision-makers, ordinary citizens); and geospatial technologies contribute to these fields, such as for wearable technology to track steps, etc.

In addition, human actors play a crucial role in the realization of such a system. They can be categorized as professionals (i.e., government institutions) and volunteers, including people both with and without relevant impairments. Since the involvement of people with relevant motor-disabilities is the most important for ensuring accessibility (having the right interpreters) and active participation in decision-making processes about policies, we elaborate on this subject further in the next section.

*3.2. Citizen Science as an Emerging Research Approach*

CitSci is a rapidly emerging research field, and it has the power to transform societies by increasing public awareness of societal problems and also by serving the SDGs. Regional and international CitSci associations originated around 2012 bring researchers from different disciplines and citizen scientists together and facilitate the communication between them. The Citizen Science Association, the European Citizen Science Association, the Australian Citizen Science Association, Citizen Science Asia, and other local networks (e.g., Citizen Science Center Zurich) are among these organizations. International Society for Photogrammetry and Remote Sensing (ISPRS) also founded a working group on the Promotion of Regional Collaboration in Citizen Science and Geospatial Technology at the XXIII ISPRS Congress in 2016 and has carried out several activities.

GIScience has been increasingly utilizing volunteer contributions, as can be seen from the Open Street Map project [95]. As emphasized by Goodchild [96,97], map making is not dominated by professionals anymore. According to Li and Shao [98], all public users are both geoinformation users and data and information providers. Volunteer contributions to geoinformation have been referred to as VGI, neogeography, geographic citizen science, crowdsourced geographic information, mashup, participatory sensing, and web mapping [99]. This process is supported by open geospatial software and data, and openness in both is likely to have an even greater impact in the future [100]. The most

commonly used term, VGI, describes the user-generated content related to geographic information [97] and involves crowdsourcing. A recent study by Brovelli et al. [101] discussed the importance of CitSci for making Digital Earth and categorized various approaches of citizen involvement, such as participatory sensing, community science, and civic science. The term civic science should also be considered, as it describes linking experts and stakeholders in planning social, economic, and environmental improvements [102]. The types of interactions between scientists and members of the public have further been investigated by Clark [103]. Here, we consider that VGI can also be intertwined with CitSci projects [104] when they comply with the ten principles of citizen science defined by ECSA [105]. The principles briefly emphasize the generation of new scientific knowledge through active participation and profit by and for both professional scientists and citizen scientists in multiple stages of the scientific process. The CitSci projects are also open access and open data by nature [105].

Several studies have shown that geospatial technologies, VGI, and CitSci promote not only scientific studies, but also public participation in data collection, planning, and policy and decision-making processes [106–108]. Related to disabilities, the typology of citizen scientists and their levels of participation can be categorized both for people with and without relevant impairments. As mentioned by Gharebaghi et al. [42], the obstacles that a person with an impairment faces can be best identified by themselves. It is also important for disabled persons to be at the center as policymakers. On the other hand, due to accessibility issues, people without a motor disability—both responsible government personnel and citizen scientists—must also be included in data collection and interpretation, as well as policy-making.

## 4. Discussion

As an example of a legal geography study, this article tried to explain how geoinformation technologies incorporated with CitSci approaches can be used to contribute to disability rights. These technologies give people with disabilities a voice in relation to their disability and involve them in making policies regarding their rights. This also helps with highlighting and combating the structural injustices that each disabled person experiences differently, according to their disability. Eventually, it could help to further develop the rights and freedoms specified in the Convention on the Rights of Persons with Disabilities and, more importantly, to realize these abstract rights and freedoms and make them applicable in real life.

This study aimed at describing the term 'disability' in the environmental and social context, and proposed a conceptual and a methodological framework for reducing inequalities caused by motor disabilities with the help of geoinformation technologies and CitSci. Geospatial tools and methods can detect environmental disablers, and assist the development of solutions for individual capabilities in a "where-when-how" framework. By applying the framework in practical applications, information on both enablers and disablers can be collected in a disability-oriented SDI at a national level, shared with the stakeholders, and suitable solutions can be developed in an evolutionary fashion. Thus, the basic rights and freedoms described in the Convention on the Rights of Persons with Disabilities can be practiced in reality by people with disabilities, such as access to justice, living independently and being included in the community, personal mobility, education, health, work and employment, and participation in political, public and cultural life and recreation leisure and sport.

Mobility studies are increasingly grappling with the new possibilities arising from geoinformation technologies, and the effect of geography on inequalities, accessibility, and mobility has already been the subject of several studies [42,52,74,109–111]. Combined with the SDG-related geoinformation research [13–17], the inequalities caused by disabilities could be better monitored and reduced in the future. However, a systems approach with a specially designed SDI is essential for providing widely applicable solutions that can serve to increase accessibility and achieve the SDGs related to reducing inequalities across countries. The proposed methodological framework suggests integrating the time dimension and the use of CitSci, sensor technologies, and machine learning methods in an SDI to

ensure a complete data scheme, interoperability, and dynamic character. One of the major contributions of this framework is providing evidence for policy development, by defining solid principles for the realization of abstract legal rights and freedoms for disabled persons.

The use of CitSci as an emerging research approach is currently the most realistic solution for increasing public awareness and would ensure the active participation of people with disabilities in decision-making and policy development. In order to reach this target, the development of suitable CitSci tools and methods that can be used for collecting and processing data in a way that is compliant with the disability-oriented SDI is also necessary.

There are also several limitations to the proposed approach concerning the vagueness and non-uniformity of the kinds of disabilities. There are so many different types of disabilities or even disputable or non-consensual disabilities (such as mental disability) that widen the research universe. As the diversity of disability types increases, the variety of problems for each one increases, too. In this regard, it may not be possible to identify the problem in the short term and find legal solutions to each disability type. Nevertheless, such efforts are needed to increase awareness, to develop new ideas and solutions, and to fight inequalities so that sustainable societies and environment can be ensured.

## 5. Conclusions and Future Work

In this article, a conceptual legal framework for describing the disability is provided by considering the relationship between law and geography, the structural injustice caused by geographies, and the United Nations' efforts on the rights of persons with disabilities (i.e., Convention on the Rights of Persons with Disabilities and SDGs). A methodological framework is also proposed with a systematic approach based on geospatial technologies and citizen science for data collection, analysis, and visualization. For ensuring interoperability at the national and international level, states should take responsibility for the development of disability-oriented SDI, which aligns with the state guarantees to the Convention as mentioned previously. When the methodological framework is put into practice, it would be possible to identify the geographical enablers and disablers for motor-disability and to develop proper and novel solutions. In addition, an increase in practical applications can support policy-making by collecting the required information. Thus, evidence can be provided to implement solid principles for the realization of abstract legal rights and freedoms for disabled persons.

The basic rights that can benefit from the output of such a system include, but are not limited to, access to justice, right to education, right to health, right to work and employment, living independently and being included in the community, personal mobility, respect for privacy, and other rights and freedoms concerning to participation in political, social and cultural life, and recreation leisure and sport, as defined in the Convention. Abstract legal rights have been defined on paper but their use in daily life is the one that will improve them. Ensuring the realization of rights in this way will also contribute to the reconsideration of the existing rights and lead to the development of new rights definitions.

The immediate future work on the proposed methodological framework involves identification of commonly observed motor-disability types; identification of geographical entities and social elements related to these types; and definition of methods and standard data types to develop the disability-oriented SDI and first applications of data collection and analysis for citizen scientists both with and without relevant disabilities. Further methods on data collection, analysis, and assistance applications can be developed after realizing these steps and the initial SDI can be improved after gaining further experience. In addition, it would be the most appropriate that the solutions are state-supported and budgeted. It is recommended to develop legal, political, and social policies with the contributions of lawyers, medical professionals, environmental and urban planners, geomatics engineers, etc.

**Author Contributions:** Conceptualization, writing-original draft preparation, writing-review & editing, N.O. and S.K.; investigation, N.O.; methodology, formal analysis, S.K. All authors have read and agreed to the published version of the manuscript.

**Funding:** This research received no external funding.

**Acknowledgments:** The authors sincerely thank to the anonymous reviewers for their useful comments and suggestions.

**Conflicts of Interest:** The authors declare no conflict of interest.

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
