# Peer review of "Improvement of Disability Rights via Geographic Information Science"

_sustainability, doi:10.3390/su12145807_

Round 1

Reviewer 1 Report

The manuscript portrays a methodological framework to improve disabilities rights combining GIS and citizen sciences.The topic of disability rights is of the utmost importance and can be really facilitated by GIS. 

  1. There is a fuzziness amongst conceptual, methodological and technological framework: 
    • Abstract: The main aim of this article is to propose a conceptual framework for the improvement of disability rights …”
    • Introduction: “A technological framework based on geospatial technologies and citizen involvement is also proposed”.
    • “Methodological Framework: A Technological Framework to Improve Disability Rights”
    • Discussion: “The proposed SDI framework suggests integrating the time dimension and the use of CitSci, sensor technologies, and machine learning methods to ensure a complete data scheme and dynamic character”.

What is the main aim of the paper? Describing a conceptual or a methodological framework? Or both? Why it is used the term “technological framework”? It is not just a methodological framework? SDI framework? Please stick to one term after reconsidering the type of the depicted framework, at the end the reader can be confused.

  1. The methodological framework is too generic, I would expect to see more details about the input data, and how the methods could contribute to have some output data. For instance, VGI, what kind of data could be used? 
  2. Are there any constraints / limitations? if so, please name them.
  3. What are your desired future developments for the presented methodological framework? 
  4. How could ML and AI contribute?
  5. What mobile apps could be used as assistant tools?

The authors approach a so significant topic combining various tools, methods but the lack of further details is a major weakness. 

Author Response

Dear Reviewer,

We are thankful for taking your time to review our manuscript and providing valuable comments. We believe that the quality of our manuscript was improved significantly after the revision. Attached please see our replies to your comments. We hope that the revisions we made would satisfy you.

Kind regards,

Authors

Reviewer 2 Report

This paper tries to engage with a very important societal issue in today’s data-intensive environment. To improve disability rights, the authors suggest utilizing geospatial technologies and citizen science.  The paper itself is timely and pressing, and provides a feasible framework to guide practitioners to help this specific minority group.  Moreover, I need to point out that this is a very unique paper that deals with a rarely-examined aspect related to the use of geospatial technology.

However, after a close reading of this paper, I found the structure or the depth of this paper can be further improved.  In its current form, I feel some of the discussion and arguments are difficult to understand and to link with the real world cases, especially in Section 3. By saying that, I hope the authors can consider adding some examples whenever discussing an abstractive idea.

  • For example, to help practitioners apply the where-where how model, could the author exemplify the model by a real-world case?

  • In line 353-55, the authors mentioned “ML” and “AI,” how specifically they can be applied? Or can the authors list one or two examples that implement the methodological framework illustrated in Figure 3?

Overall, I suggest a major revision to allow the authors to exemplify the methods which are proposed in section 3 with real-world cases

Author Response

(The authors gave the same response as above.)

Reviewer 3 Report

The manuscript in question has several limitations:

The relationship between GIS and other fields including disabilities and/or law has long been established and cited in the literature. In fact, several studies can be found in literature discussing walkability and accessibility using GIS techniques. Thus, I don’t think that there is a need for any new article to show this relationship.

The authors are also not consistent in their writing and are not focused on the primary objective of the study (improvement of disability rights), often including examples/references from various disciplines instead. This is confusing for the reader.

The study does not include any practical example showing data that use of GIS has improved the rights of people having disabilities.

No conclusion drawn of the research.

I am not sure if I should treat it as a review article or a research article as it does not meet the essential requirements of both.

Author Response

(The authors gave the same response as above.)

Reviewer 4 Report

General

The paper presents an interesting work that handles how Geographic Information Science can improve disabled people's rights. Although it is very theoretical work, after reading the title, I have missed some maps or some geographic representations.

Expressions

Please, be cautious with the expression "and so forth" because it looks informal, in lines 47, 327, 369.

American/British English

Please, you have to choose American or British English and be coherent with the election. The clearer example is found in lines 254 (emphasised) and 255 (emphasized).

Among others:

L 154: medicallised and indivisualised.

L 156: emphasized

L 254: analised

L 280: analyse

L 369: harmonized

L 418: visualisation

Citation and References

As it appears in the Instructions for Authors:

For embedded citations in the text with pagination, use both parentheses and brackets to indicate the reference number and page numbers; for example [5] (p. 10). or [6] (pp. 101–105).

Now it is shown as

Line31: ", and more [1], p.5";

Similar in other lines, such as in 33, 37, 113.

L424: "Foody et al" should be replaced with "Foody et al."

Regarding the Reference section:

I have noticed that many papers are not correctly described as they are incomplete. The most common mistake is lack of volume or issue. For example,

Incomplete (as it is): ISPRS Int. J. Geo-Inf. 2019, 8, 515

Complete: ISPRS Int. J. Geo-Inf. 2019, 8(11), 515

Even better: ISPRS Int. J. Geo-Inf. 2019, 8(11), 515, doi:10.3390/ijgi8110515

Author Response

Dear Reviewer,

We are grateful for taking your time to review our manuscript and providing valuable comments. We revised our manuscript based on your suggestions. Attached please see our replies to your comments.

Kind regards,

Authors

Reviewer 5 Report

After a detailed review of this paper, these are my suggestions:

  1. I am not sure if the guidelines of this journal allow to add references in this style ([1], p.5.)
  2. You should reduce the number of references in the whole manuscript. Almost 120 is a overwhelming number. But of course, you should reduce in the parts where they are not important at all, and to maintain or even increase in the parts where are important. This is my perspective.
    1. For example, in the Introduction, paragraph 2: Line 42-61. “Sustainable development and reducing inequalities have become primary goals of all nations. (…)”. You should reduce most of the references there because they do not add any special value.
    2. The same is happening with the references 98-101. They are the same one. BTW, the reference 101, which comes from a Facebook page does not enough scientific.
    3. The Citizen Science Association. Available online: https://www.citizenscience.org/ (Accessed on 23 April 2020).
    4. European Citizen Science Association. Available online: https://ecsa.citizen-science.net/ (Accessed on 23 April 2020).
    5. Australian Citizen Science Association. Available online: https://citizenscience.org.au/ (Accessed on 23 April 2020).
    6. Citizen Science Asia. Available online: https://www.facebook.com/CitSciAsia/ (Accessed on 23 April 2020).
    7. Citizen Science Center Zurich. Available online: https://citizenscience.ch/de/ (Accessed on 23 April 2020)

  1. You should not add a reference every time you add an institution number, because in this way, the number of references will be too extremely high for a research paper. For example here “The Citizen Science Association [98], the European Citizen Science Association [99], Australian Citizen Science Association [100], Citizen Science Asia [101], and other local networks (e.g. Citizen Science Center Zurich, [102]) are among these organizations.”

  1. The most interesting part for me of your manuscript is related to the technical part. It is precisely here where I am expecting a well managed literature review. The authors say: “As an example, mobile apps can help disabled persons get information about specific public places concerning their disability problems. A specially designed app can be useful to collect data about a specific building or a park place depending on other disabled persons’ experiences. In this regard they would know whether/not or how to get that place safely. However, contributions of non disabled people or automated methods are also needed, since they can provide data about places with low or no accessibility for disabled people. On the other hand, crowdsourced data from smartphones, e.g. street images, or IoT (e.g. street cameras) with geolocation information can be used to detect potential physical obstacles automatically and possibly in real-time, which can contribute to the information repository as depicted in Figure 3. Various studies for street object detection from mono or stereo images or lidar data can be found in recent literature (e.g. [81-85]). VGI can as well be used for the similar purposes, i.e. spatial data provision or accessibility validation, with the direct or indirect inputs from humans (e.g. please see [86-88]).

Considering these requirements, ML and artificial intelligence (AI) methods deserve special attention. These methods can support GIS for automated processing and analysis. An AI supported  GIS platform, which employs web and mobile technologies, spatial databases, statistical and image processing software libraries, and visualisation tools to assist the volunteers for spatial data collection  was developed by [89].”

The authors introduce many different technologies and it is ok. However, the contextualization of the use of these technologies in Improvement of Disability Rights is missing. For example, I can give you some advices from my field of expertise. Some studies analyze the urban metrics and classify the land uses within urban areas. In the case of urban metrics (street width, sidewalks width, relationship between street and height of buildings, etc), it has a special importance for designing a city for disabled people. Some papers are really interesting for analyzing it such as “Using street based metrics to characterize urban typologies”.

One second focus with a great interest and great applicability is related to the sentiment analysis and how the city is perceived for disabled citizens. In this case, naturalistic studies are fundamental. These studies analyze the real behavior of people in real-world conditions. Thus, it helps to reply to questions about how disable people perceive the city and why they take some decisions. These kind of studies add a focus clear to the presented issue in your manuscript. Papers like “GIS mapping of driving behavior based on naturalistic driving data” or “The influence of road familiarity on distracted driving activities and driving operation using naturalistic driving study data” can be interesting to refer in your case.  

Author Response

(The authors gave the same response as above.)

Round 2

Reviewer 1 Report

The authors improved significantly the quality of the manuscript and addressed all my questions.

Author Response

Dear Reviewer,

We would like to thank again for taking your time to review our manuscript and providing your valuable inputs, which definitely improved the quality of our work.

Kind regards,

Authors

Reviewer 2 Report

In this revision, the authors have updated the structure by adding a conclusion section, and provided particular examples to make the argument easier understandable. This manuscript is much improved by incorporating these two major edits.

One thing I encourage the authors to consider: the concept “Citizen Science” was posed as one of the two dependent approaches. After reading Section 3.2, I see most of the reviewed literature is geography or GIScience relevant. It reminds me perhaps “VGI” is more precise or relevant opposite to “citizen science.” Is that possible to put both geospatial technology and citizen science, or more precisely “VGI”, under a bigger umbrella term? By doing so, the paper will present a more holistic approach instead of two loosely related approaches.

Author Response

Dear Reviewer,

We are very happy to have your valuable comments and suggestions on our manuscript. After your suggestion, we have decided to revise the title of the manuscript as following:

“Improvement of Disability Rights via Geographic Information Science”

Minor revisions also exist in the Abstract.

We agree that the majority of the content is related to GIScience, but the use of citizen science is also important since it would ensure the active participation of people with disabilities in decision-making and policy development. Therefore we wanted to emphasize citizen science in the initial title. However, we understand your point of view and the revised title can be more suitable for the manuscript.

We hope that the revisions we made would satisfy you.

Kind regards,

Authors

Reviewer 3 Report

Geospatial and other latest technologies are extensively used to identify inequalities and to implement a solution. Thus, it is not a new conceptual framework. It is already being practiced in many developed countries. It is up to the developing countries to follow it or not.

Yes there is always a need for improvement as technology is changing at a fast pace. However, one has to identify the shortcomings of existing models and practices to propose improvements. I believe the manuscript is lacking in that.

Author Response

Dear Reviewer,

We are thankful for your valuable comments on our manuscript and useful insights on the subject. We would like to provide further clarifications regarding our view, as there might be some misunderstanding or lack of clear explanations.

We have made revisions in the Conclusions and Future Work Section. In addition, we have made further clarifications to your concerns in the attached file. 

We hope that the revisions we made would satisfy you.

Kind regards,

Authors

Round 3

Reviewer 3 Report

Please see my detailed comments on the attached file.

Author Response

Dear Reviewer,

We have revised our manuscript further based on the comments and suggestions of the Reviewers. We would like to thank you for taking your valuable time to review our manuscript and your contributions.

Kind regards,

Authors